

# Acteoside alleviates UUO-induced inflammation and fibrosis by regulating the HMGN1/TLR4/TREM1 signaling pathway

Yan Mao[1,2], Jiali Yu[1,2,3], Jingjing Da[1,2,3], Fuxun Yu[1,2,3] and Yan Zha[1,2]

[1] School of Medicine, Guizhou University, Guiyang, Guizhou, China
[2] Department of Nephrology, Guizhou Provincial People's Hospital, Guiyang, Guizhou, China
[3] NHC Key Laboratory of Pulmonary Immunological Disease, Guizhou Provincial People's Hospital, Guiyang, Guizhou, China

## ABSTRACT

**Purpose**. Acteoside (Act), a phenylethanoid compound that was first isolated from mullein, has been widely used for the investigation of anti-inflammatory and anti-fibrotic effect. However, the mechanism of Act against unilateral ureteral obstruction (UUO)-mediated renal injury is largely unknown. Therefore, this study aimed to explore the effects of Act on UUO rats and possible mechanisms.

**Methods**. A total of 20 Sprague-Dawley (SD) rats were divided randomly into three groups ($n \geq 6$): (i) sham-operated group (Sham); (ii) UUO group (UUO+Saline); and (iii) UUO + Act 40 mg/kg/day, (UUO+Act); Continuous gavage administration for 2 weeks postoperatively, while the rats in Sham and UUO+saline groups were given equal amounts of saline. All rats were sacrificed after 14 days, the urine and blood samples were collected for biochemical analysis, the renal tissues were collected for pathological staining and immunohistochemistry. Correlations between individual proteins were analyzed by Pearson correlation analysis.

**Results**. The results of renal function indexes and histopathological staining showed that Act could improve renal function by reducing serum creatinine, blood urea nitrogen and urine protein at the same time, Act could alleviate renal inflammation and fibrosis. In addition, the results of immunohistochemistry showed that Act could reduce the expression of inflammation and kidney injury-related proteins F4/80, Mcp-1, KIM-1 proteins, as well as the expression of fibrosis-related protein $\alpha$-SMA and $\beta$-catenin. More importantly, Act can also reduce the expression of HMGN1, TLR4 and TREM-1 proteins.

**Conclusion**. These data demonstrate that Act can ameliorate UUO-induced renal inflammation and fibrosis in rats probably through triggering HMGN1/TLR4/TREM-1 pathway.

Corresponding author
Yan Zha, zhayan72@126.com

## INTRODUCTION

Urinary tract obstruction (UTO) is a common clinical disease and an important cause of renal injury. The incidence of UTO in China is about 6.1%−6.4% (*Wu et al., 2020*; *Yang et*

*al., 2020*), usually caused by urinary tract stones, benign prostatic hyperplasia, and pelvic or ureteral tumors. UTO may lead to serious consequences such as tubulo-interstitial fibrosis, inflammation and eventually develop into obstructive nephropathy, which is one of the important causes of chronic kidney disease (*Chevalier, Forbes & Thornhill, 2009*; *Better et al., 1973*). As increased intratubular hydrostatic pressure and secondary ischemia progress, renal injury such as tubular cell death, renal inflammation and fibrosis (*Wen et al., 1999*) occurs which leads to a chronic kidney disease (CKD) and eventually to end-stage renal disease (ESRD) (*Martínez-Klimova et al., 2019*). According to the latest hospitalization data, chronic kidney disease accounts for about 4.84%, of which about 16% with obstructive nephropathy. Especially in southern China, the proportion rises to 28.7% (*Zhang et al., 2020a*). Thus, the incidence of UTO has an important impact on the incidence of chronic kidney disease.

UTO can be treated by surgery, but the renal fibrosis cannot completely be eliminated. Because of the renal fibrosis plays an important role in predicting outcome, anti-fibrosis is still the main treatment direction and the treatment of renal fibrosis is an urgent problem to be solved (*Stevens, 2018*).

In research, the UUO model is often used to study the etiology and pathogenesis of obstructive nephropathy caused by UTO, which is most commonly ligation of the left ureter using a silk (*Ucero et al., 2014*). As a model of obstructive nephropathy, UUO offers unique advantages, such as the absence of exogenous toxins, the lack of a uremic environment, and observe the contralateral kidney as a control (*Aranda-Rivera et al., 2021*). In the UUO rat model, the metabolism and hemodynamics of the kidney were altered. On the one hand, when the increased intratubular pressure and the regulation of the nervous system work together will cause secondary renal vasoconstriction, reducing the blood flow of the renal blood vessels. These events will result in a shunt of blood to the kidneys that may eventually lead to renal ischemia. At the same time, changes of the pressure in the renal tubules may cause mechanical stretching of cells, resulting in cell damage or even necrosis. At this time, the normal function of the renal tubules is affected, leading to the occurrence of renal tubular diseases (*Ucero et al., 2010*; *Bai et al., 2020*). On the other hand, the interstitial cells of the renal tubules are also affected. Due to changes in renal hemodynamics, macrophages in the interstitial tubules may be activated to release inflammation-related cytokines such as interleukin-6 (IL-6), tumor necrosis factor-α (TNF-α), and monocyte chemoattractant protein-1 (MCP-1) or activate the expression of related genes like the aggregation of vascular smooth muscle actin (α-SMA) and β-catenin, eventually leading to inflammation and fibrosis (*Bai et al., 2021*). The severity of the disease is closely related to the progression of renal failure, therefore, in the state of UUO, a series of pathological changes will appear in the kidney and these changes play a key role in the process of promoting irreversible renal damage (*Zhou et al., 2020a*; *Kido et al., 2017*; *Ma et al., 2016*; *Wang et al., 2018*; *You et al., 2019*; *Lee, Kim & Choi, 2015*). *Tammaro et al. (2013)* reported that the expression of triggering receptor expressed on myeloid cells-1 (TREM-1) positive cells may be closely related to inflammation and fibrosis in the kidney, and more interestingly, its role may be achieved by amplifying the inflammatory signal of TLR4. The TLR4 pathway is a classic inflammatory pathway, and its role in renal inflammation and

fibrosis has been reported in many articles (*Wang et al., 2020*; *Sari et al., 2021*; *Zhang et al., 2015*). Mobility nucleosome-binding protein 1 (High-mobility group nucleosome-binding protein 1, HMGN1) is a natural TLR4 activator (*Jiang et al., 2021*). In recent years, *Yang et al. (2012)* have published two research results, suggesting that HMGN1 can also secrete to extracellular and play the biological activity of "alarm. Studies showed that HMGN1 can recruit and activate immune cells through the TLR4 pathway (*Ang et al., 2018*).

At present, Chinese medicine has also been widely reported for the treatment of UUO rats due to its unique advantages such as extensive pharmacological effects and long-lasting efficacy (*Lin et al., 2018*). Research by *Zhang et al. (2020b)* validated that Asiatic acid attenuated renal injury, oxidative stress, and fibrosis in UUO rats by using targeted GC-MS-based metabolomics; reports have also shown that Puerarin Alleviates can alleviated UUO-mediated fibrosis and inflammation through triggering TGF β1/Smads and NF-κB p-65/STA T-3 pathways (*Wang et al., 2021*).

Acteoside (Act, also known as Verbascoside or Kusagin.), (β-(3,4-dihydroxyphenylethyl)-O-α-L-rhamnopyr-anosyl-(1 →3)- β-D-(4-O-caffeoyl)-glucopyranoside) (Fig. 1A), is a type of phenylpropanoid glycoside, which named after its initial isolation from the mullein (*Alipieva et al., 2014*). It can be found in many medicinal plant families such as Verbenaceae, Lamiaceae, and Scrophulariaceae. Studies have reported that Act has a certain pharmacological effect in anti-inflammatory (*Akbay et al., 2002*). Researchers have demonstrated that Act have an improving effect on alcohol-induced liver function damage by inhibiting the NF-kb pathway, while also inhibiting the development of liver fibrosis (*Venditti et al., 2012*). Subsequently, *Khullar et al. (2019)* also found that Act could also be used as a potential drug of choice for primary hepatocellular carcinoma in 2020.

However, there are scarce reports has been conducted on the use of Act in UUO, based on some previous studies on HMGN1 protein by our team, the purpose of this study was to explore whether Act can alleviate renal inflammation and fibrosis, and also investigate whether this process occurred through triggering HMGN1/TLR4/TREM-1 pathway.

## MATERIALS & METHODS

### Animals

Male Sprague-Dawley (SD) rats (6–8 weeks old, body weight 200 ± 10 g) were purchased from Chongqing Teng Xin Biotechnology Co. Ltd (Chongqing, China) and used for this study. The rats were maintained with standard rat chow and tap water at room temperature (21 °C ± 2 °C) and stable humidity under a light/dark cycle of 12/12 h. The research was approved by the ethics committee of Guizhou People's Hospital (Approval Number: (2017)070). Twenty SD rats were randomly assigned to the sham surgery group (Sham, $n \geq 6$ per group), the UUO group (UUO+Saline, $n \geq 6$ per group), and the UUO with Act treatment group (UUO+Act, $n \geq 6$ per group) (*Faul et al., 2007*).

### Unilateral ureteral obstruction surgery

UUO is a model widely used to study obstructive nephropathy, enabling the elucidation of the cellular and molecular events involved in obstructive renal injury. In the modeling
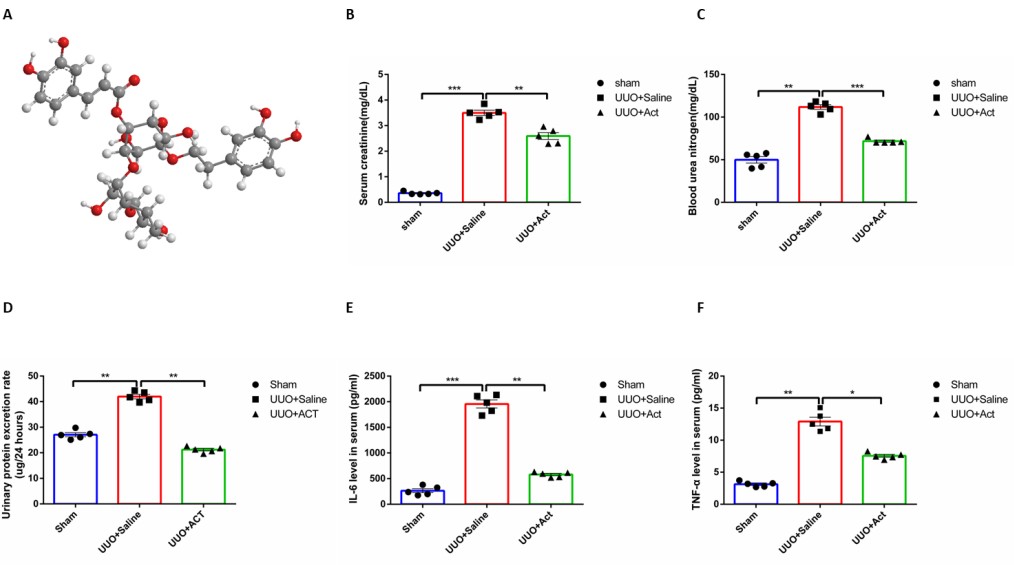

**Figure 1  Act alleviated the renal function parameters and systemic inflammation in unilateral ob-struction (UUO) rat models.** (A) Structural of Acteoside (Act); (B, C, D) Act decreases the levels of serum creatinine (Scr), blood urea nitrogen (BUN), urinary protein excretion rate in UUO groups. (E, F) The serum concentration of IL-6 and TNF-$\alpha$ in UUO rats were determined using enzyme-linked immunosorbent assay (ELISA) kits. All data are represented as mean $\pm$ SEM, $n = 5$ rats per group. $*p < 0.05$, $**p < 0.001$, $***p < 0.0001$.

experiment of UUO, the left ureter is usually ligated with silk thread. It is also necessary to maintain the animal on a surface heated to body temperature.

UUO surgery in this study was performed using an established protocol as described previously (*Gu et al., 2019*; *Strutz & Zeisberg, 2006*). Briefly, rats are randomly divided into three groups, to minimize pain in rats and after anesthesia with 4% pentobarbital sodium (50 mg/kg, Sigma, Merck, Germany), the rats were kept warm on a mat, and then operated on. The rat abdomen was opened to expose the left ureter and then ligated with 4-0 silk. (a) Sham surgery group: opened the abdomen of rats to expose the left kidney, without the ligation of the ureter; (b) UUO+Saline group: opened rats' abdomen to expose the left kidney and ligated the left ureter with 4-0 silk. (c) UUO+ Act group: 14 days after UUO surgery in rats, received intragastrical gavage of Act (40 mg/kg (body weight)/day, dissolved in normal saline, pH 7.3, 0.154 mol/L) (*He et al., 2011*). Rats in the Sham surgery group and UUO+Saline group received normal saline. Recorded the weight of each animal before administration. On 14 days after UUO, each rat was placed in a metabolic cage one day in advance, 24-hour urine is collected, followed by intraperitoneal injection of 4% pentobarbital sodium for euthanasia, blood is collected, and tissue is perfused with heparin-containing saline (*Cheng et al., 2010*), kidneys was kept in 10% formalin solution for histological examination.

## Determination of blood urea nitrogen and serum creatinine

On the 14th day of continuous dosing, 24-hour urine was collected, and after centrifugation at 3,500 rpm for 15 min at 4 °C, urine protein was detected using a protein estimation kit (Bio-Assay Systems, Hayward, USA). Blood samples were centrifuged at 3,500 rpm for 15 min at 4 °C to obtain serum, and blood urea nitrogen and serum creatinine concentrations were determined using a urea assay kit and a creatinine assay kit (Bio-Assay Systems, Hayward, USA), respectively. Three biochemical indicators were measured according to the protocol provided by the manufacturer.

## Histopathological evaluations

Renal tissue was fixed in 10% formalin solution and embedded in paraffin, then sliced using a rotary slicer (RM2245, Wetzlar, Germany). Sections (2 μm thickness) were stained with hematoxylin and eosin (HE, BOSTER Biological Technology Co.Ltd, Wuhan, China) and Masson's Trichrome (Sigma-Aldrich, Merck KGaA, Darmstadt, Germany) following proposed protocols. The scoring standard was described in the article by *Wu et al. (2022)*. According to its standards: 10 fields of view covering randomly the whole kidney section for each slice were quantified (every field contains 100 renal tubules). One score for obvious expansion; one score for flat tubular cells; one score for vacuolization of epithelial cells in the renal tubules; one score for renal tubular necrosis without protein cast; one or two scores according to the severity of brush border injury or desquamation; two scores for protein casts. Average the score of these 10 fields and the average score was taken as the finally score for subsequent statistics. The results of Masson's trichrome staining were expressed by image-pro-plus V6.0 to calculate the percentage of renal interstitial collagen deposition in the total renal interstitial area. 10 visual fields were selected for each sample and the average value was taken as the percentage of fibrosis for each sample.

## Immunohistochemistry

Tissue section of kidney (4 μm thickness) were deparaffinized in xylene and rehydrated in an alcohol gradient. Then sections were incubated for 10 min in endogenous peroxidase blocking solution (BOSTER, Wuhan, China) at room temperature in dark to block endogenous enzymes. Washed with 0.01 mol/L phosphate-buffered saline (PBS) and antigen retrieved in high pressure with citrate buffer (PH6, Beijing Zhongshan Golden Bridge Biotechnology, China) for 2 min, Sections were blocked of nonspecific by using 0.01 mol/L PBS containing 5%(w/v) bovine serum albumin (Solarbio, Beijing, China) at room temperature for 30 min, incubate the slices with anti-F4/80(1:1000 dilution, Protein-tech, Wuhan, China), anti-HMGN1(1:200 dilution, Bio-World, Nanjing, China), anti-MCP1(1:200 dilution, beyotime, Shanghai, China), anti-KIM-1(1:50 dilution, Abcam, Cambridge, UK), anti-TLR4 (1:50 dilution, HUABIO, Hangzhou, China), anti α-SMA (1:100 dilution, Abcam, Cambridge, UK), anti-TREM-1(1:200 dilution, BOSTER, Wuhan, China) antibodies at 4 °C overnight. The antibody was diluted with 0.01 mol/L PBS containing 1% [w/v] bovine serum albumin. Washed with 0.01 mol/L PBS and Incubated secondary antibodies, (goat anti-rabbit IgG (1:200 dilution); Abcam, Cambridge, UK) for 90 min at room temperature and for 30 min at 37 °C in thermostat-controlled

water-bath. Finally, the signal was visualized by using a 3, 3′-diaminobenzidine (DAB) kit (Beijing Zhongshan Jinqiao Biotechnology, Beijing, China). Then counterstained with hematoxylin, dehydrated with gradient alcohol and xylene and sealed with neutral balsam.

## Statistical analyses

All statistics data were expressed as mean $\pm$ SEM, analyzed using GraphPad Prism 6.0 (GraphPad Software, San Diego, CA, USA). Two-tailed Student's $t$-test was used to compare between two groups; statistical significance between groups was calculated using an ANOVA test; the correlation between two variables by using Pearson coefficient to analyze. $P$-values less than 0.05 ($p < 0.05$) were considered to be statistically significant.

## RESULTS

### Influence of Acteoside on renal function and release of inflammatory factors in serum in UUO rat

Serum creatinine (Scr), and blood urea nitrogen (BUN) are classic indicators for evaluating renal function. To assess whether Act had an improving effect on renal function, we measured Scr and BUN levels in three groups, as well as urine protein in each group after 14 days. The experimental results have shown that Scr and BUN levels were significantly higher in the UUO group than in the control group, while decreased after treatment with Act (Figs. 1B–1D). Comparing with the control group, the Urine protein in the UUO group was significantly reduced, and the situation was improved after intervention with Act.

Serum levels of inflammatory factors can reflect systemic inflammation to a certain extent, in order to investigate the inflammatory changes in each group, we examined the serum level of inflammatory factors IL-6 and TNF-α in three groups. As depicted in Figs. 1E and 1F, the inflammatory factors of serum in the UUO group were significantly higher than in the Sham group, and the serum level of inflammatory factors decreased after Act treatment.

### Influence of Acteoside on renal inflammation and injury in UUO rat

To assess the pathological changes in rat kidneys, this study performed HE staining on kidney slices. The staining results of HE showed that rats in the UUO model had pathological changes in kidney, such as renal tubular cells necrosis fell off and filled with a tube-shaped protein, which could be improved after Act treatment (Fig. 2A). The score is based on the degree of kidney damage, as can be seen from Fig. 2C, the renal injury score in the UUO+Saline group was higher than that in the Sham group, whereas the renal injury score in the UUO+Act was lower than that in the UUO+Saline group.

To explore the effects of Act on kidney injury, the study examined the expression of Mcp-1 (monocyte chemotactic protein 1) and kidney injury molecule (KIM)-1 in each group, is an immunoglobulin super-family protein that is markedly upregulated in the proximal tubule of the injured kidney (*Han et al., 2002*). Meanwhile, studies have shown that the accumulation and activation of macrophages in diabetic kidneys can cause glomerular and tubular damage, albuminuria, elevated plasma creatinine, renal fibrosis and expression of chemokines (*Calle & Hotter, 2020*; *Klessens et al., 2017*). We investigatd

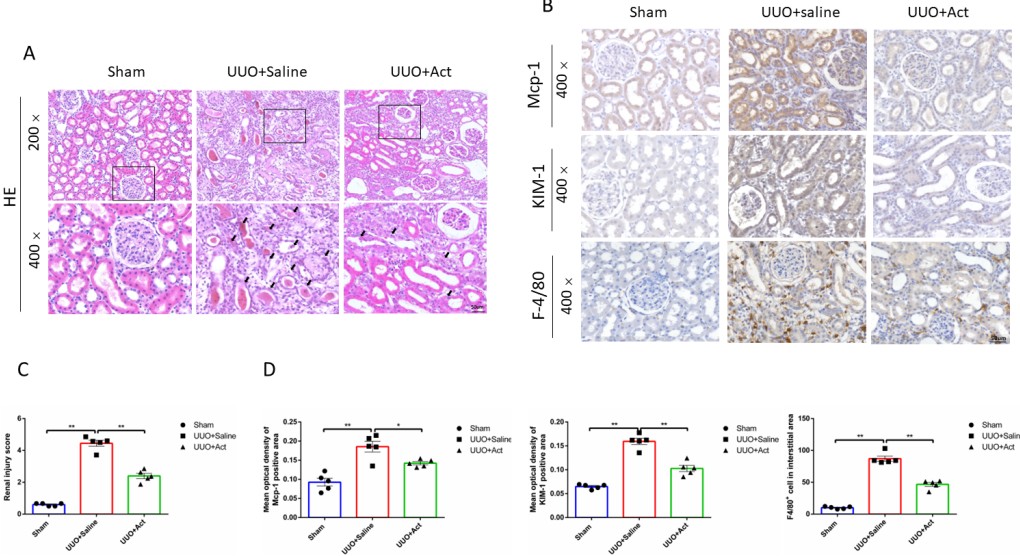

**Figure 2  Act attenuates the renal inflammation and injury in unilateral obstruction (UUO) rat models.** (A) Kidney sections from all groups were stained with H&E for morphology changing, " → " point to renal tubular cells necrosis fall off and filling with a tube-shaped protein; (B) immunohistochemical stain of Mcp-1, KIM-1 and F-4/80 in the kidney tissue of UUO rats ($n = 5$). Original magnification = 400 ×. Scale bars. 50 um. All data are represented as mean ± SD; (C) Renal injury score was analyzed based on H&E staining; (D) quantification of the levels of Mcp-1, KIM-1 and F-4/80 proteins. all values are presented as mean ± SEM. *$p < 0.05$, **$p < 0.001$.

whether Act can reduce the inflammatory response induced by UUO injury, so this study detected the expression of typical macrophages (F4/80+ positive cells). As shown by the results of immunohistochemistry found that the expressions of Mcp-1, KIM-1 proteins and the number of F4/80+ positive cells in the UUO+Saline group were higher than in the Sham group, and UUO+Act group was lower than those in the UUO+Saline group (Figs. 2B and 2D).

## Influence of Acteoside on renal fibrosis in UUO rat

In addition, renal tubular fibrosis is often thought to be the result of a failed wound-healing of kidney damage, which is also an important cause leading to ESRD (*Zhou et al., 2020b*). In our studies, the Masson staining results showed almost no renal fibrosis in Sham group, while the UUO+Saline group showed severe renal fibrosis, this renal fibrosis can also be ameliorated after Act treatment and semi-quantitative as percentage of the fibrotic area (Figs. 3A and 3C).

To assess whether Act can improve renal fibrosis at the protein level, we performed immunohistochemical analysis of kidney tissue. The main feature of renal interstitial fibrosis is the accumulation of α-SMA and β-catenin protein (*Li et al., 2019*; *Zeng, Xiao & Sun, 2019*). The results of this study showed that levels of α-SMA and β-catenin protein were significantly enhanced in UUO+Saline group, while Acteoside may mitigate those changes (Figs. 3B and 3D).

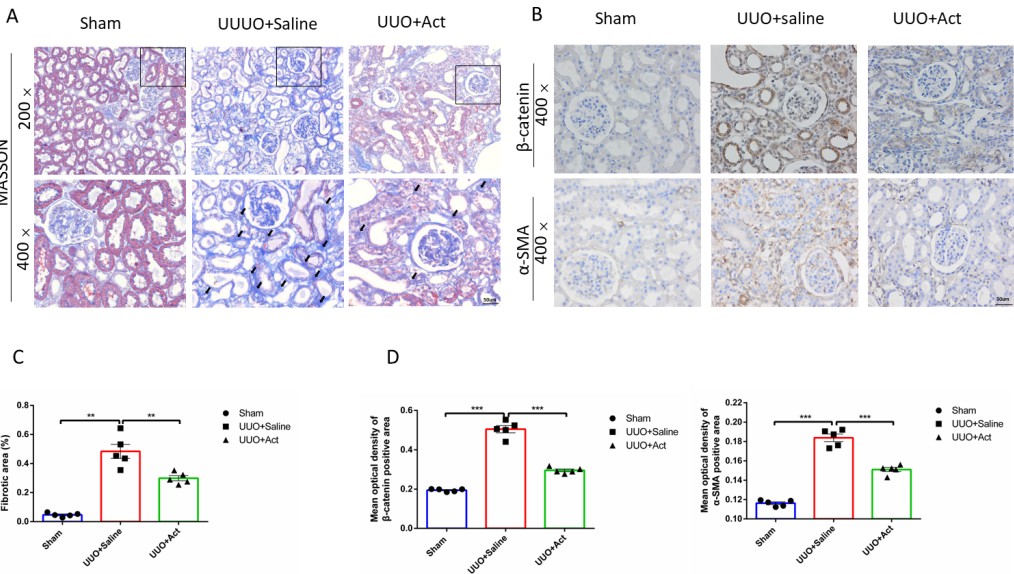

**Figure 3 Act reduced the accumulation of collagen fibers and the expression of key proteins involved in UUO-induced renal fibrosis.** (A) Masson trichrome staining for collagen fibers (B) immunohistochemical stain of $\beta$-catenin and $\alpha$-SMA in the kidney tissue of UUO rats ($n = 5$). Original magnification $=400 \times$. Scale bars. 50um. All data are represented as mean ± SEM; (C) Percentage of the fibrotic area by Masson's trichrome staining. (D) Quantification of the levels of $\beta$-catenin and $\alpha$-SMA proteins, all values are presented as mean ± SEM. **$p < 0.001$, ***$p < 0.0001$.

## Influence of Acteoside on the expression of HMGN1, TLR4, TREM-1 proteins in UUO rat

HMGN1 is a nucleosome-binding protein. In recent years, studies have found that it can activate TLR4 and activate immune cells (*Jiang et al., 2021*; *González-Romero, Eirín-López & Ausió, 2015*). Therefore, to investigate whether HMGN1 regulate the occurrence of renal inflammation and the process of fibrosis, this study used immunohistochemistry to detect the levels of HMGN1 and TLR4 in rat kidney slices. As illustrated in Fig. 4, compared with the sham group, the expression of HMGN1 ang TLR4 proteins were increased in UUO+Saline group, and the expression levels of HMGN1 and TLR4 proteins in UUO+Act were lower than those in the UUO+Saline group.

TREM-1 signaling pathways have been reported to play key roles in mediating pro-inflammatory and pro-fibrosis processes in obstructive nephropathy (*Xianyuan et al., 2019*; *Liu et al., 2020*). This study also detected the expression of TREM-1 protein in different groups of kidney tissues. The results of immunohistochemistry have shown that levels of TREM-1 protein were significantly enhanced in UUO mice, while UUO+Act group was lower than those in the UUO+Saline group (Fig. 4).

## Significant correlations between HMGN1 and related effector proteins

This study also focused on the associations among proteins or positive stain cells, using Pearson analysis to assess them. As can be observed in Fig. 5, HGMN1 protein level positively correlated with macrophage-related markers (F4/80+) cell counts, Mcp-1 and

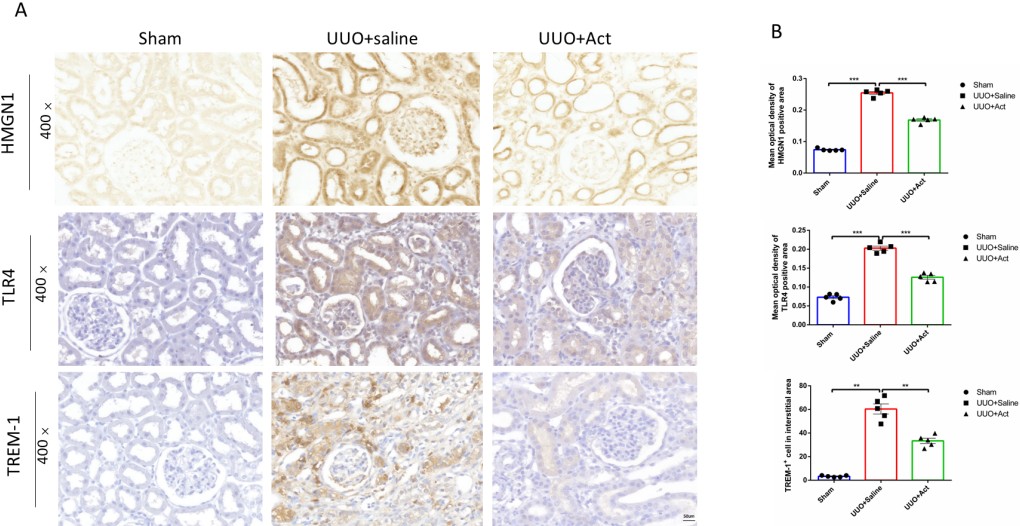

**Figure 4 HMGN1, TLR4, and TREM-1 protein levels in the kidney of different experimental groups.**
(A) Immunohistochemical stain of HMGN1, TLR4 and TREM-1 in the kidney tissue of UUO rats ($n = 5$).
Original magnification = 400 ×. Scale bars. 50 um. (B) Quantification of the levels of HMGN1, TLR4 and
TREM-1 proteins. all values are presented as mean ± SEM. **$p < 0.001$, ***$p < 0.0001$.

KIM-1 respectively ($P < 0.0001$, r2 = 0.9694, $n = 15$; $p < 0.0001$, r2 = 0.8116, $n = 15$; $p < 0.0001$, r2 = 0.8685, $n = 15$, respectively). At the same time, the level of HMGN1 protein was also positively correlated with the level of α-SMA and β-catenin proteins ($p < 0.0001$, r2 = 0.9451, $n = 15$; $P < 0.0001$, r2 = 0.9268, $n = 15$). Subsequently, the analysis also found a significant positive correlation between HMGN1 and TLR4 proteins ($p < 0.0001$, r2 = 0.9482, $n = 15$). It is worth noting that there is a significant positive correlation between TLR4 and TREM-1 respectively ($p < 0.0001$, r2 = 0.9071, $n = 15$, respectively).

## DISCUSSION

Act is a component of natural medicine, accumulating evidence supported that it has many positive effects in anti-inflammatory (*Viswanatha et al., 2021*). In this study, we demonstrated that Act effectively ameliorate the renal inflammation and fibrosis.

The UUO model is commonly used to mimic the obstructive nephropathy. In rats with the UUO, renal interstitial macrophage infiltration, excessive interstitial extracellular matrix deposition, and renal fibrosis (*Ucero et al., 2010*). The results of HE staining and Masson's trichrome staining in this study can confirm that in the kidneys of UUO rats, compared with the Sham group, a series of obvious changes appeared in UUO+saline group. Such as renal tubular dilatation and obvious casts in the renal tubules, the renal tubular epithelial cells fell off, the numbers of renal interstitial macrophage increased, and the deposition of collagen fibrils increased. Most pathologic changes mainly cause the renal tubule damage, which could contribute to excretion level rise in the Scr, BUN and urinary protein. Meanwhile, the inflammatory factors in body's serum are also increased. Significantly, all the indicators decreased in UUO+Act group. Therefore, we made a

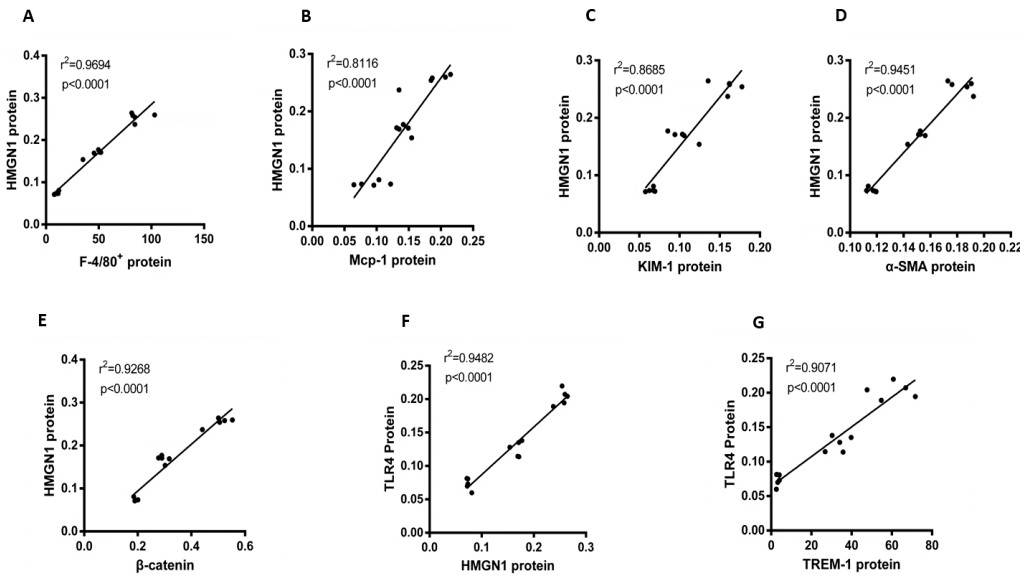

**Figure 5** **Correlation analysis of HMGN1 protein with the key proteins involved in UUO-induced renal inflammation and fibrosis.** (A) HMGN1 protein had a positive relation with F4/80+ cells. (B) HMGN1 protein was positively associated with Mcp-1 protein. (C) HMGN1 protein was positively related to KIM-1 protein. (D) HMGN1 protein was positively associated with $\alpha$-SMA protein. (E) HMGN1 protein was positively associated with $\beta$-catenin protein. (F) TLR4 protein and HMGN1 protein had a positive relation. (G) TLR4 protein was positively associated with TREM-1 protein.

preliminary inference that Act could protect the kidney by improving renal function and reducing inflammation in the body.

Moreover, in order to explore the effects of Act on kidney injury, the study examined the expression of proteins associated with inflammation and fibrosis, we found that in UUO+saline group, the level of the kidney injury marker (KIM-1), the macrophage marker (F4/80), and the levels of fibrosis-related markers ($\alpha$-SMA and $\beta$-Catenin) were significantly higher than Sham group, indicating that inflammation and fibrosis have presented in UUO rat kidneys. Similarly, the expression level of these proteins was decreased by using Act treatment. According to the above research results, we further infer that act has ameliorating effects on renal inflammation and fibrosis.

For the treatment of non-infectious inflammation, there is currently a lack of definite therapeutic drugs. Therefore, we expect to explore new therapeutic methods from the perspective of regulating molecular pathways in the body. The three families, HMGA, HMGB, and HMGN, constitute the high mobility protein superfamily. HMGB1 has been reported to activate the body's innate immunity (*Jiang et al., 2021*; *Yang et al., 2012*; *Ang et al., 2018*). Recent studies have shown that HMGN1 is an alarmin protein that induces an inflammatory or immune response by activating dendritic cells; notably, their found that the effect of HMGN1 was more efficient and more stable than that of HMGB1. Based on the previous research on the mechanism of HMGN1 in inflammation, we have shown that HMGN1 can induce inflammatory or immune responses by triggering TLR4-dependent signaling pathways in a type 1 diabetes mice (*Yu et al., 2018*). Therefore, we are very
concerned about whether HMGN1 protein is involved in a series of pathological changes in UUO rats.

TREM-1 is a member of the immunoglobulin superfamily, the research of *Nguyen et al. (2020)* shows that TLR4 activated the TREM-1 on the surface of macrophages and promote the release of inflammatory factors. Moreover, studies have proved that TREM-1 can be used as a diagnostic marker in kidney diseases, including obstructive kidney disease. And in their experiments, finding deplete TREM-1 ameliorates the UUO renal pathology (*Lo et al., 2014*). In a report by *Nguyen-Lefebvre et al. (2018)* TREM-1 on Kupffer cells can release a large amount of chemokine Mcp-1 to activate hepatic stellate cells (HSCs), and finally promote the occurrence of liver fibrosis. Additionally, *Feng et al. (2021)* suggested that in mouse macrophages, TREM-1 could enhance pro-inflammatory immune responses by inducing mononuclear macrophages to produce chemokine MCP-1 and TNF-α. According to the above studies, we assumed that a series of inflammatory responses and the development of fibrosis in the kidneys of UUO rats may occur by triggering HMGN1/TLR4/TREM-1-dependent signaling pathways.

According to this assumption, we detected the expression of HMGN1, TLR4 and TREM -1 proteins in the kidneys of each group by immunohistochemistry. Consistent with the expected results, the levels of HMGN1, TLR4 and TREM -1 proteins in the UUO+Saline group was significantly higher than that in the Sham group, and the expression of these proteins was decreased after Act treatment. In order to further verify our assumption, we did a series of correlation analysis. We found that there is a significant correlation between HMGN1 protein and inflammation, fibrosis molecules and TLR4 protein. Meanwhile, there is also a significant correlation between TLR4 protein and TREM-1 protein. Based on the above results, we speculate that HMGN1 protein may trigger TLR4 protein, which subsequently initiates the TREM-1 signaling pathway, which ultimately leads to kidney inflammation and fibrosis in UUO rats, and that Act can protect kidneys by improving inflammation and fibrosis.

Structurally, the caffeoyl moiety in the chemical structure of Act shows a great affinity to phospholipid membranes with negative charge; mechanically, the phenyl rings of Act can form hydrogen bonds with other residues, changing the hydrogen-bond network in the structure and combine to the target region with a low binding energy (*Xiao, Ren & Wu, 2022*). Based on the above characteristics, Act can better pass through the cell membrane and finally exert its pharmacological effects. More importantly, in terms of drug safety, Act does not show genotoxic activity and phototoxicity in the body (*Henn et al., 2019*). These advantages suggested that Act has a promising therapeutic drug candidate for obstructive nephropathy.

Unfortunately, our study still has some limitations. This study has not yet clarified the specific mechanism by which HMGN1 protein participates in Act to improve renal inflammation and fibrosis, and has not conducted in-depth research on the active center of Act, so we will do further investigation in the follow-up study.

## CONCLUSIONS

In conclusion, our study demonstrated that Act could mitigate UUO-mediated renal inflammation and fibrosis. The underlying mechanism possibly is related to the HMGN1/TLR4/TREM-1 signaling pathway.

## ACKNOWLEDGEMENTS

This work was done at the NHC Key Laboratory of Pulmonary Immunological Disease in Guizhou Provincial People's Hospital, China. The authors are grateful to the staff in the laboratory for their technical assistance.

### Funding

This work was supported by grants from the Special Fund for Basic Scientific Research Operating of Central Public Welfare Research Institutes, the Chinese Academy of Medical Sciences (2019PT320003), the Guizhou High-Level Innovative Talents Program (QKHPTRC(2018)5636-2), the Guizhou Clinical Research Center for Kidney Disease (QKHPTRC[2020]2201), the Science and Technology Fund project of Guizhou Provincial Health Commission in 2021 (gzwkj2021-136), the Youth Fund of Guizhou Provincial People's Hospital in 2021 (GZSYQN[2021]12), and the Basic Research Plan of Guizhou Province in 2022 (Natural Science Project) (QKH-ZK[2022] General 265). The funders had no role in study design, data collection and analysis, decision to publish, or preparation of the manuscript.

### Grant Disclosures

The following grant information was disclosed by the authors:
Special Fund for Basic Scientific Research Operating of Central Public Welfare Research Institutes.
Chinese Academy of Medical Sciences: 2019PT320003.
Guizhou High-Level Innovative Talents Program: QKHPTRC(2018)5636-2.
Guizhou Clinical Research Center for Kidney Disease: QKHPTRC[2020]2201.
Science and Technology Fund project of Guizhou Provincial Health Commission: gzwkj2021-136.
Basic Research Plan of Guizhou Province in 2022 (Natural Science Project): QKH-ZK[2022] General 265.

### Competing Interests

The authors declare there are no competing interests.

### Author Contributions

- Yan Mao conceived and designed the experiments, performed the experiments, analyzed the data, prepared figures and/or tables, authored or reviewed drafts of the article, and approved the final draft.

- Jiali Yu conceived and designed the experiments, performed the experiments, analyzed the data, prepared figures and/or tables, authored or reviewed drafts of the article, and approved the final draft.
- Jingjing Da conceived and designed the experiments, prepared figures and/or tables, and approved the final draft.
- Fuxun Yu conceived and designed the experiments, authored or reviewed drafts of the article, and approved the final draft.
- Yan Zha conceived and designed the experiments, prepared figures and/or tables, authored or reviewed drafts of the article, and approved the final draft.

## Animal Ethics

The following information was supplied relating to ethical approvals (i.e., approving body and any reference numbers):

The animal study on rats began after approval from the ethical committee of the Guizhou Provincial People's Hospital [(2017)070] and all the experimental procedures were performed according to the National Institutes of Health Guide for the Care and Use of Laboratory Animals and China animal welfare legislation.

## Data Availability

The raw data is available in the Supplemental Files.

## Supplemental Information

Supplemental information for this article can be found online at http://dx.doi.org/10.7717/peerj.14765#supplemental-information.

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
