# Peer review of "Acteoside alleviates UUO-induced inflammation and fibrosis by regulating the HMGN1/TLR4/TREM1 signaling pathway"

_PeerJ, doi:10.7717/peerj.14765_

## Round 0.1 · original submission · Minor Revisions

The authors are asked to correct the typo errors. The manuscript is well-written and organized. The results are well discussed. The manuscript would be considered for publication after minor corrections.

Reviewer 1 ·

Basic reporting

no comment

Experimental design

no comment

Validity of the findings

In this manuscript, authors reported that Acteoside (Act) can improve renal functions by regulating various proteins and Act also involves in UUO mediated renal inflammation and fibrosis possibly via HMGN1/TLR4 pathway. This is very interesting study.
However, I have few suggestions/comments regarding this manuscript before it goes to publication.
1. Authors did immunohistochemistry of each group rat kidney tissues and found that HMGN1, TLR4 and TREM1 levels were improvised after Act treatment. Is there any mechanistic evidence to prove HMGN1/TLR1/TERM1 signalling in UUO induced inflammation and fibrosis?
2. HMGN1 shows significant positive correlation with MCP-1, KIM-1, F4/80, α-SMA, β-catenin, TLR4 and TREM1. What does it tells exactly to support this manuscript conclusion.
3. To support their immunohistochemistry data, Is there any other way authors checked levels of all these proteins in 3 different group animals? (Ex: Western blotting..etc)

·

Basic reporting

Please correct the following typos:
Line 34 - urine "protein"
Line 275 - 277 - Therefore, we made a preliminary "inference that Act" could protect the kidney by improving renal function and "reducing" inflammation in the body
Line 331-333 - "In" conclusion, our study demonstrated that Act could mitigate UUO-mediated renal 332 inflammation and fibrosis. The underlying mechanism possibly "is" related to the 333 HMGN1/TLR4/TREM-1 signaling pathway.

Experimental design

no comment

Validity of the findings

no comment

Additional comments

Summary:
Initially discovered from mullein, acteoside (Act) is a phenylethanoid molecule that has been extensively used to study its anti-inflammatory and anti-fibrotic effects. However, it is mostly unknown how Act protects against unilateral ureteral obstruction (UUO)-mediated renal damage. Therefore, the purpose of this study was to investigate any potential processes behind Act's effects on UUO rats.

A total of 20 Sprague-Dawley (SD) rats were randomly assigned to one of three groups (ng6): the sham-operated group (Sham); the UUO group (UUO+Saline); and the UUO + Act group (UUO+Act). Continuous gavage administration was used for two weeks following surgery, with the rats in the Sham and UUO+saline groups receiving the same amount of saline. After 14 days, all rats were slaughtered, and kidney tissues were removed for pathological staining and immunohistochemistry. Blood and urine samples were also taken for biochemical examination. The Pearson correlation analysis was used to examine correlations among distinct proteins.
According to renal function indices and histological stains, Act can lessen renal inflammation and fibrosis while also improving renal function by lowering serum creatinine, blood urea nitrogen, and urine protein. Additionally, the results of immunohistochemistry demonstrated that Act might decrease the expression of fibrosis-related proteins 3-SMA and 3-catenin, as well as proteins associated with inflammation and kidney injury such as F4/80, Mcp-1, and KIM-1. More significantly, Act can also lower the expression of the proteins HMGN1, TLR4, and TREM-1.

These findings show that Act can reduce renal inflammation and fibrosis brought on by UUO in rats, most likely by activating the HMGN1/TLR4/TREM-1 pathway.

The results are well discussed and presented in this manuscipt

---

## Round 0.2 · accepted · Accept

The manuscript is ready for publication. The authors have addressed the minor errors pointed out by the reviewers.